# Complex Analysis of Micronutrient Levels and Bone Mineral Density in Patients with Different Types of Osteogenesis Imperfecta

**DOI:** 10.3390/diagnostics15030250

**Published:** 2025-01-22

**Authors:** Diana Valeeva, Karina Akhiiarova, Ildar Minniakhmetov, Natalia Mokrysheva, Rita Khusainova, Anton Tyurin

**Affiliations:** 1Internal Medicine and Clinical Psychology Department, Bashkir State Medical University, 450008 Ufa, Russia; diana2537@yandex.ru (D.V.); liciadesu@gmail.com (K.A.); ritakh@mail.ru (R.K.); 2Endocrinology Research Centre, Dmitriya Ulianova Street, 11, 117036 Moscow, Russia; minniakhmetov.ildar@endocrincentr.ru (I.M.); mokrisheva.natalia@endocrincentr.ru (N.M.)

**Keywords:** BMD, osteogenesis imperfecta, micronutrients, calcium, magnesium, copper, zinc, phosphorus

## Abstract

**Background:** Osteogenesis imperfecta (OI) is a rare monogenic connective tissue disorder characterized by fragility of bones and recurrent fractures. In addition to the hereditary component, there are a number of factors that influence the course of the disease, the contribution of which is poorly understood, in particular the levels of micronutrients. **Methods:** A cross-sectional study was conducted involving 45 with OI and 45 healthy individuals. The concentrations of micronutrients (calcium, copper, inorganic phosphorus, zinc, and magnesium) and bone mineral density (BMD) were evaluated in all the participants. **Results:** The concentrations of micronutrients in all the groups were within the reference values. In the OI overall, magnesium and copper were elevated, and phosphorus and zinc were lower. Type I exhibited higher concentrations of magnesium and copper and the lowest phosphorus; type III was associated with lower zinc, type IV with lower calcium and higher copper, and type V with the lowest phosphorus. OI overall was associated with lower BMD values. A correlational analysis in the OI group showed that the number of fractures correlated with BMD in absolute values but not with the Z-score. **Conclusions:** The obtained data emphasize the importance of the levels of micronutrients in the pathogenesis of connective tissue diseases, in particular OI. As in the results of previous studies, the levels of micronutrients were within the population norm, which probably requires the development of individual criteria for the content of substances in this category of patients.

## 1. Introduction

Osteogenesis imperfecta (OI) is a rare connective tissue disorder characterized by reduced bone density, fragility of bones, and recurrent fractures [1]. The prevalence of this condition in the population ranges from 1:15,000 to 1:20,000 [2].

This disorder is also associated with extraskeletal manifestations, such as blue sclerae, dentinogenesis imperfecta, joint hypermobility, and hearing loss [3]. Additionally, OI is a phenotypically heterogeneous condition with a wide range of clinical manifestations, ranging from a moderate increase in fracture frequency over a lifetime to severe, progressive skeletal deformities due to recurrent fractures and pathological mineralization of bone tissue [4]. The clinical severity of the disease is classified along a spectrum from perinatal lethal (OI type II) to severe (OI type III), moderate (OI type IV), and mild (OI type I), as initially classified by Sillence and Rimoin [5]. Recurrent “brittle bone” fractures are accompanied by pain, prolonged rehabilitation, and disability, significantly decreasing the patient’s quality of life. The multisystem involvement of connective tissue underlies the severity of this condition.

OI is associated with pathological changes in collagen structure, leading to the “fragility” of bone tissue and visceral manifestations. Over 1500 dominant mutations have been identified in the *COL1A1* and *COL1A2* genes in OI patients. These mutations result in either a structural or quantitative deficiency of type I collagen. Additionally, mutations have been observed in several genes involved in extracellular post-translational modification of collagen (*CRTAP*, *LEPRE1*, and *PPIB*), collagen folding and intracellular transport (*SERPINH1* and *FKBP10*), ossification or mineralization (*SERPINF1*), and osteoblast development (*WNT1*, *CREB3L1*, and *SP7*) [6], which are critical mechanisms in bone tissue metabolism. However, comprehensive understanding of the entire pathogenesis of OI is still lacking. Several studies have indicated that micronutrients are essential for the growth and development of the skeleton, as they interact with the micronutrients in bones and participate in normal bone metabolism [7]. It is known that certain micronutrients, such as phosphorus, calcium, copper, zinc, magnesium, vitamin D, and proteins, influence bone structure and metabolism, thereby preventing skeletal disorders [8,9,10,11,12]. However, there has been insufficient research examining these elements in patients with OI, predominantly focusing on pediatric cohorts. The precise roles of micronutrients in the pathogenesis of OI remain largely unknown, and understanding the relationship between these elements and areal bone mineral density (BMD) may provide valuable insights for the diagnosis and intervention of OI. Moreover, the search for potentially new pathogenic targets for pharmacological therapy of OI represents a promising avenue for future research.

The aim of this study was to analyze the levels of micronutrients and the mineral density of bone tissue in adult patients with OI.

## 2. Materials and Methods

A case-control study was conducted involving 45 patients diagnosed with OI and 45 individuals of comparable gender and age without a history of monogenic connective tissue disorders, endocrine disorders, rheumatological and autoimmune diseases, traumas, or skeletal fractures. All the patients received inpatient treatment at the Bashkir State Medical University clinic from 2021 to 2024. The study procedure was thoroughly and clearly communicated to all the participants, and informed consent was obtained voluntarily. This research was conducted in accordance with the Declaration of Helsinki (1964) and received approval from the local ethics committee of Bashkir State Medical University (protocol No. 8 dated 23 October 2021). The diagnosis of OI was established based on clinical presentation and molecular genetic diagnostics at the connective tissue research center of the BSMU clinic by a multidisciplinary team that consisted of a rheumatologist, a general practitioner, and a medical geneticist. In the first stage, the serum concentrations of micronutrients—calcium, copper, inorganic phosphorus, zinc, and magnesium—were assessed in all the subjects, with the biochemical parameters evaluated using spectrophotometry.

The evaluation of areal bone mineral density (BMD) was conducted using dual-energy X-ray absorptiometry (DXA) on a Lunar Prodigy Advance device (GE HealthCare, USA), with the calculation of absolute values and Z-scores.

### 2.1. Control Group

The inclusion criteria were a young age from 18 to 35 years. The exclusion criteria included the presence of hereditary forms of connective tissue disorders (Marfan syndrome, Ehlers–Danlos syndrome, and osteogenesis imperfecta), metabolic bone diseases, autoimmune diseases, a history of trauma, chronic kidney disease stage 3b or higher, the presence of endocrine diseases, conditions associated with micronutrient metabolism, pregnancy and lactation, taking medications that affect bone mineral density, and refusal to participate.

### 2.2. Statistical Data Processing

The statistical data processing was performed using GraphPad Prism 10.4.0. (GraphPad Software, San Diego, CA, USA) and Excel 2021, Version 2109 (Microsoft Office 2021, Microsoft Corporation, Washington, DC, USA). The assessment of the normality of the distribution of variables was conducted using the Kolmogorov–Smirnov–Lilliefors test. The pairwise intergroup comparisons were carried out using Student’s *t*-test and the Mann–Whitney U test, with a significance level set at *p* ≤ 0.05. The association between variables was evaluated using the Pearson and Spearman correlation coefficients. The assessment of the strength of the relationship was conducted according to the Cheddock scale.

A statistical analysis comparing two independent groups on parameters such as magnesium and phosphorus was conducted using Student’s *t*-test. In the comparison of other parameters (age; height; weight; BMI; number of fractures; levels of calcium, zinc, and copper; BMD; and Z-score), the hypothesis of normality in distribution was rejected, and the Mann–Whitney U test was used.

## 3. Results

The cohort of patients with OI included 20 (44.4%) males and 25 (55.6%) females, with a median age of 32.0 years. Mutations were identified in 34 (75.5%) patients, confirming the molecular diagnosis. Twenty-three patients had mutations in the *COL1A1* or *COL1A2* genes associated with types I, III, and IV of OI according to the D. Sillence classification. Type I of OI is characterized by autosomal dominant inheritance, blue sclera, hearing impairment, osteopenia, and non-deforming fractures. The patients with type III OI exhibited bluish or grayish sclera and pronounced skeletal deformities with a high number of fractures, with inheritance being autosomal recessive. Type IV OI (autosomal dominant) is characterized by wide clinical variability, dental enamel pathology, hearing impairment, and primarily deformities of long tubular bones. Three patients were found to have a mutation in the *IFITM5* gene, which is characteristic of type V OI, presenting with hyperplastic bone calluses in fracture areas [13]. Three patients were identified with mutations in the *P3H1*, *LEPRE*, *CRTAP*, *SERPINF1*, *PLOD2* (Bruck syndrome), *TGFB1*, and *SGMS2* genes (Calvarial Doughnut Lesions with Bone Fragility) [14,15]. The patients with these mutations in non-collagenous genes had type I OI. The number of fractures experienced throughout life by the patients with OI varied from 1 to 170 (median 15.0 [10.0; 30.0]). Additionally, the patients with types III and IV of OI displayed higher body mass index (BMI) values, reaching statistical significance when compared to the control group. Statistically significant differences in the presence of joint hypermobility were observed only in type I OI, whereas the prevalence of dental enamel pathology was higher across all types of the condition. The characteristics of the studied groups are presented in Table 1.

The control group included 17 (37.8%) men and 28 (62.2%) women, aged 20 to 39 years (median age of 27.0 years), comparable in gender composition and age to the OI patient group.

In the first stage, an assessment of the serum concentrations of micronutrients was conducted. When comparing the concentrations of micronutrients in the OI group as a whole and the control group, the concentrations of calcium, magnesium, inorganic phosphorus, zinc, and copper in both groups were within the reference values. However, in the OI group overall, the serum concentrations of magnesium and copper were significantly elevated (*p* < 0.05 and *p* < 0.05, respectively), while the phosphorus (*p* < 0.05) and zinc (*p* = 0.045) levels were lower compared to the control group. The calcium levels did not differ significantly between the two groups (Table 2).

A subsequent analysis was conducted on the concentration of micronutrients in the serum of patients with different types of OI in comparison with a control group. When the studied groups were categorized according to OI type, the levels of micronutrients remained within the reference range. The patients with type I OI exhibited significantly higher concentrations of magnesium (*p* < 0.05) and copper (*p* = 0.039), while the phosphorus levels (*p* < 0.05) were lower compared to the control group. Type III OI was associated with lower zinc levels (*p* < 0.05) and, despite a higher incidence of fractures within the group, did not differ from the controls in terms of calcium and phosphorus concentrations. The patients with type IV OI demonstrated the lowest zinc values compared to other OI types and the control group; however, these differences did not reach statistical significance. Additionally, the type IV OI group exhibited lower calcium concentrations (*p* < 0.05) and higher copper levels (*p* < 0.05). Type V OI was associated with the lowest serum phosphorus levels among the other OI types, with the differences achieving statistical significance when compared to the control group (*p* < 0.05) (Table 3).

The next step was to assess the BMD levels in both absolute values and Z-score values in the OI and control groups. Overall, OI was associated with lower BMD values. The lowest BMD values were observed for type III OI (0.84 g/sm^2^ and −2.15, respectively), while the highest absolute values were found in type I (1.03 g/sm^2^). The obtained value of BMD for type I OI is close to that of the control when compared to other types of OI, yet they still differ significantly, with a high level of significance (1.182 g/sm^2^ and 1.03 g/sm^2^; *p* < 0.05). The highest values of the Z-criterion were found for the control group (Z-score 0.71). When comparing the BMD values with the control group, all the types of OI exhibited lower absolute values, reaching statistical significance (Figure 1 and Table 4).

In the analysis of BMD in relative terms (Z-scores), types I, III, and V of OI were significantly associated with decreased BMD when compared to the control group.

A correlational analysis conducted in the groups with OI as a whole and in the control group revealed a moderate inverse correlation between copper and zinc (r = −0.323 and *p* = 0.03). In the group with OI overall, the concentrations of zinc and magnesium exhibited a direct moderate significant correlation (r = 0.322 and *p* = 0.034). Additionally, in the OI group, the number of fractures correlated with BMD in absolute values but not with the Z-score. The results are presented in Figure 2.

## 4. Discussion

Despite the fact that the serum concentrations of micronutrients in our study were within the reference ranges, differences were noted when comparing various types of OI with the control group. In light of the limited data regarding the influence of micronutrients on bone metabolism in patients with OI, the authors propose a generalized scheme (Figure 3) based on the literature. For instance, Karita K. et al. (2001) found elevated zinc levels in the nails of patients with OI compared to the control group [16]. Our findings revealed lower zinc concentrations for type III OI; however, this assessment was conducted in serum rather than in nails. Zinc is one of the most critical micronutrients involved in nearly all metabolic processes within the body. It is the only metal that is a component of all six classes of enzymes (oxidoreductases, transferases, hydrolases, lyases, isomerases, and ligases). Approximately 50% of this micronutrient is stored in muscle tissue, 30% in bones, and 20% in other organs and tissues [17]. Zinc not only contributes to bone tissue but also plays a role in the synthesis of the collagen matrix, mineralization, and remodeling of bone [18].

Zinc may be associated with the inhibition of both osteoblastogenesis and osteoclastogenesis through the suppression of the expression of the transcription factor Runt-related 2 (RUNX2), which is induced by the Wnt/β-catenin pathway [19].

Furthermore, Zn significantly reduces bone resorption by mediating the inhibition of the RANKL/OPG pathway [20]. Additionally, it has been revealed that zinc can prevent osteoblast apoptosis induced by oxidative stress by initiating a series of cascade reactions that lead to a reduction in cellular oxidation, inhibition of cytochrome C release, and decreased phosphorylation of P38 and JNK, which are involved in cell death signaling [21]. A meta-analysis conducted in 2014 by researchers from China demonstrated that higher serum levels of zinc, copper, and iron are associated with a lower risk of osteoporosis [22].

The level of zinc in the blood was positively associated with lumbar BMD (total T-score) among postmenopausal women in Turkey. Suzuki T. et al. (2015) suggested that zinc deficiency may indirectly influence BMD by reducing intestinal calcium absorption and enhancing the effects of bone resorption by parathyroid hormone [23] due to elevated levels of parathyroid hormone [24]. It is worth noting that the use of these micronutrient reference values may not be entirely appropriate in the case of rare diseases such as OI, and therefore, the development of specific micronutrient standards for patients with this disease may be necessary.

Types I and IV OI were associated with higher concentrations of copper when compared to the control group. Copper serves as a cofactor for the enzyme lysyl oxidase, which forms intra- and intermolecular cross-links in collagen, thereby playing a crucial role in the cross-linking of lysine residues in collagen and elastin, enhancing the mechanical strength of fibrils. A reduction in the copper concentration within the body, which leads to an insufficient number of cross-links, results in growth disorders and osteogenesis, as well as the brittleness of bone tissue. However, it has been shown that the relationship between copper status and the risk of osteoporosis is nonlinear. An analysis of NHANES data (2011–2014) indicated a lower total BMD of the femur and femoral neck, with subjects exhibiting the highest copper concentrations having a fourfold increased risk of fracture, while an increase of 10 µg/dL in serum copper levels heightened the risk of fractures in men. Moreover, elevated serum copper concentrations lead to a reduction in bone size and density in C57 mice [25].

Excess copper also leads to the formation of a considerable number of free radicals, which induce lipid peroxidation and disrupt bone metabolism, resulting in a reduction in the cortical layer and decreased bone strength [26]. Our findings are consistent with previous research results. However, during the correlational analysis, copper levels were found to be unrelated to BMD and the number of fractures, but they exhibited a moderate negative correlation with serum zinc concentrations in both the control and OI groups, which may be explained by the antagonism of these trace elements in the process of absorption in the gastrointestinal tract. Furthermore, a moderate direct correlation between zinc and magnesium was noted, albeit only in the OI group. Our study revealed higher serum magnesium levels for type I OI compared to the control group. Research on the impact of magnesium on the development of osteoporosis is limited; however, the role of magnesium in the metabolism of connective tissue overall, and specifically bone tissue, is challenging to overstate. Magnesium is an important micronutrient involved in many physiological processes within the body. More than 50% of magnesium is concentrated in bone tissue, dentin, and dental enamel.

The transport of this micronutrient within the body is regulated by antidiuretic peptide, glucagon, calcitonin, parathyroid hormone, and insulin. Elevated levels of magnesium may also lead to disturbances in the mineralization processes of bone tissue. High levels of magnesium in bone tissue inhibit the formation of hydroxyapatite crystals by competing with calcium and binding with pyrophosphate to form insoluble salts [27]. The anti-osteoporotic effect of magnesium is mediated through the enhanced regulation of the BMP-2/6 and Wnt/β-catenin signaling pathways, as well as the activation of PI3K/Akt and ERK and the stimulation of GSK3β phosphorylation. RANKL-dependent osteoclastogenesis is also inhibited by magnesium, which is closely associated with the production of osteoprotegerin [28]. However, even for magnesium, which possesses a relatively wide therapeutic window, high doses have been shown to inhibit osteoblastogenesis and increase the differentiation of osteoclasts. The literature regarding the impact of serum magnesium levels on bone mineral density is contradictory. Increased magnesium intake in postmenopausal women has been associated with a higher incidence of wrist fractures [29]. Nevertheless, a significant correlation between magnesium consumption and BMD of the hip has been demonstrated in a cross-sectional study involving men aged 69–97 years [30].

I type OI was associated with lower serum phosphorus levels. Phosphorus is the second primary component of bone tissue, following calcium, that forms the inorganic bone matrix (calcium hydroxyapatite and octacalcium phosphate), the metabolism of which is regulated by parathyroid hormone and vitamin D. Phosphorus deficiency is relatively rare and, in most cases, is caused by impaired phosphorus reabsorption in the kidneys, potentially leading to the formation of unmineralized osteoid. Conversely, excessive phosphorus intake, particularly in cases of calcium deficiency in the body, contributes to the development of secondary hyperparathyroidism and active bone resorption; high phosphorus consumption does not have detrimental effects when the Ca:P ratio is normal. Research conducted by scientists in Japan indicated that high phosphorus levels inhibit the differentiation and activity of osteoclasts [31], while a Brazilian osteoporosis study revealed a 9% increase in fracture risk for every 100 mg/day increase in phosphorus intake among patients with low calcium consumption [32].

In our study, lower serum calcium levels were associated exclusively with type IV OI and did not correlate with areal bone mineral density or fracture incidence in both the control and OI groups. The majority of the body’s calcium is deposited in bones in the form of hydroxyapatite, which is responsible for mineralization. A constant concentration of calcium is essential for various biological functions; therefore, in cases of hypocalcemia, there is an increased release of parathyroid hormone, leading to enhanced calcium reabsorption and the production of calcitriol. This process subsequently stimulates the resorption of bone tissue by osteoclasts, resulting in the release of calcium and phosphorus into the bloodstream [30]. High concentrations of calcium in the cytosol modify the cytoskeleton of osteoclasts. This may be mediated by calcium signaling molecules expressed in podosomes, which alter cell adhesion and reduce bone resorption. However, elevated levels of extracellular calcium stimulate the formation of osteoclast-like cells and the resorption of bone by mature osteoclasts [33].

In the case of OI, the increased brittleness of bones is attributed to changes in the properties of bone tissue, as well as to low bone density [34]. Bone tissue, classified as a type of connective tissue, comprises bone cells and an extracellular matrix. The extracellular matrix consists of a ground substance, predominantly made up of inorganic material, which imparts solidity and brittleness, alongside an organic component that provides plasticity. The strength of bone is conferred by collagen fibers [35]. The decreased collagen content in the bones of patients with OI is associated with the thinning of the cortical layer and a lower trabecular index as observed in histomorphometry for type I OI, which is caused by a quantitative defect in collagen without anomalies in the primary structure or post-translational modifications of type I collagen. In contrast, types II, III, and IV exhibit defects in the primary structure of the alpha chains of collagen, as well as in the post-translational modifications of the collagen triple helix [36].

Both types of collagen defects are associated with an increase in osteoblasts and osteoclasts, which sustains a high level of bone tissue metabolism and a constant ineffective remodeling process during which new matrix formation occurs, which is also defective, similar to the old matrix in normal mineralization processes [37]. In our study, all types of OI were associated with lower absolute values of BMD and the Z-scores were reduced in all types of OI except for type IV. According to the literature, low bone mineral density may be more prevalent in OI type III compared to OI type I or IV, which aligns with our findings. It is noteworthy that our study employed the DXA method, which has several limitations associated with its use for assessing and monitoring osteopenia/osteoporosis, particularly in OI. Primarily, DXA is a two-dimensional method and can only measure areal BMD, which may not necessarily diminish in OI [38] due to hypermineralization [39].

Additionally, the data may be distorted due to limb deformations resulting from recurring fractures, the shorter stature of patients with OI, the presence of metal constructs, and the inability to position patients due to contractures [40]. Furthermore, according to the results of our study, the number of fractures exhibited a moderate direct correlation with absolute values of BMD, but not with relative values in the OI group, indicating the need for new approaches in the interpretation of BMD values obtained through DXA for this patient cohort.

Currently, bisphosphonate therapy and surgery utilizing telescopic rods are considered the gold standard for the treatment and correction of bone deformities in OI [41]. The use of micronutrients has been proposed as a therapeutic alternative to enhance bone healing following the implantation of metal constructs. Researchers from Brazil investigated the impact of adding micronutrients (calcium, magnesium, and zinc) on the restoration of bone tissue around implants in mice. No statistically significant differences were observed between the groups receiving placebo and those receiving micronutrient supplementation based on computed tomography results and the measurement of the torque required to disrupt the bone–implant interface [42]. The effects of other micronutrients on the recovery of bone tissue following surgical intervention warrant further investigation.

Our study has several limitations. Only 45 patients from the OI group were included in this study of the rare condition, necessitating an expansion of the sample size. Additionally, our research does not account for comorbidities in patients with the condition. Furthermore, our study is of the case-control type, which does not allow for a comprehensive assessment of the causal relationships between the components of the pathogenesis and the levels of micronutrients. It is also important to note that only the serum levels of micronutrients were assessed; to obtain a complete picture, it is necessary to evaluate the concentrations of these micronutrients directly in bone tissue.

## 5. Conclusions

Despite serum concentrations of micronutrients remaining within the reference values, differences were observed when comparing various types of OI and the control group. In type I OI, higher concentrations of magnesium and copper were noted. Type V OI exhibited the lowest serum phosphorus levels. Type III OI was associated with lower zinc levels, and despite a higher number of fractures in this group, there was no significant difference compared to the control group in terms of the calcium and phosphorus concentrations. Lower serum calcium levels were found exclusively in type IV OI and did not correlate with DXA-derived BMD or the number of fractures in either the control or OI groups. The number of fractures demonstrated a moderate positive correlation with absolute BMD values but not with the Z-score in the OI group.

The obtained data emphasize the importance of the levels of micronutrients in the pathogenesis of connective tissue diseases, in particular OI. As in the results of previous studies, the levels of micronutrients were within the population norm, which probably requires the development of individual criteria for the content of substances in this category of patients. Also, in the patients with OI, absolute BMD values are a more sensitive marker for assessing the risk of fractures compared to the level of the Z-criterion, which are based on population standards.

## Figures and Tables

**Figure 1 diagnostics-15-00250-f001:**
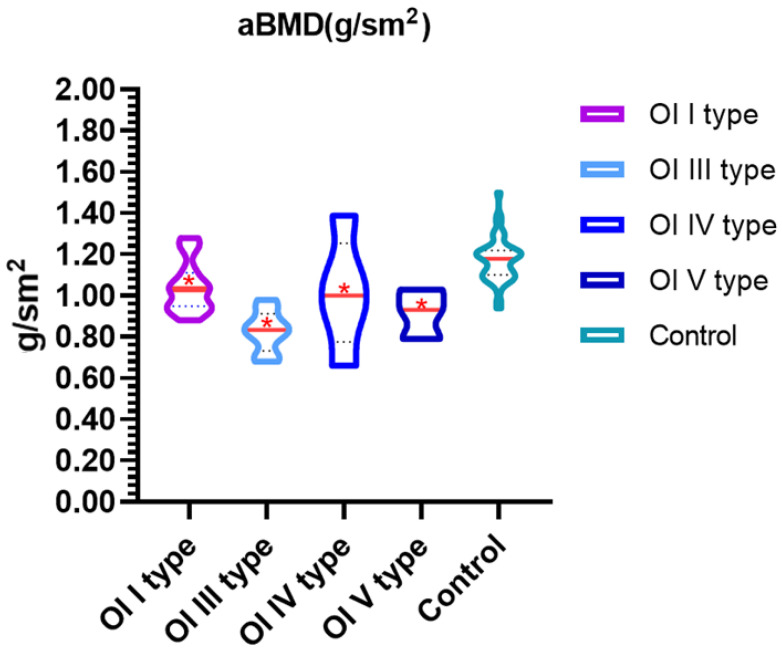
Comparative analysis of BMD in OI and control groups. In the violin plot, the horizontal line indicates the median BMD, while the dashed lines represent the 25th and 75th percentiles, respectively. * *p*-value < 0.001.

**Figure 2 diagnostics-15-00250-f002:**
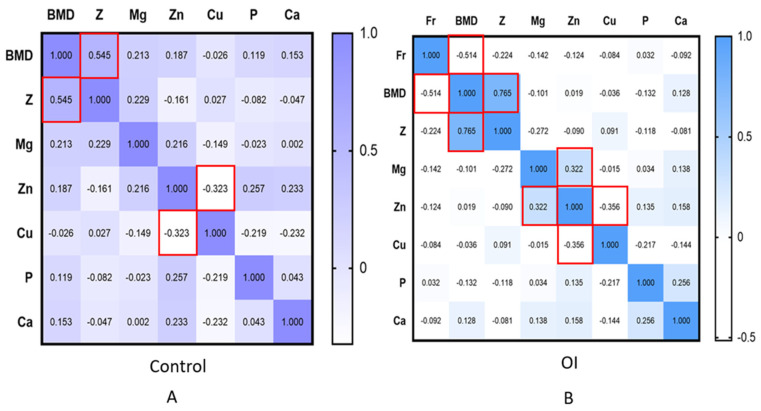
Results of the correlation analysis (R—Spearman). Significant correlation relationships are highlighted in red (*p* < 0.05). BMD—bone mineral density, Z—Z-score, Fr—fractures, Mg—magnesium, Zn—zinc, Cu—copper, P—phosphorus, and Ca—calcium. In the control group (**A**), moderate negative correlations were found between copper and zinc, and moderate positive correlations were observed for the Z-criterion and BMD; in the group with osteogenesis imperfecta (**B**), moderate negative correlations were identified between zinc and copper, the number of fractures, and BMD, as well as direct moderate correlations between zinc and magnesium.

**Figure 3 diagnostics-15-00250-f003:**
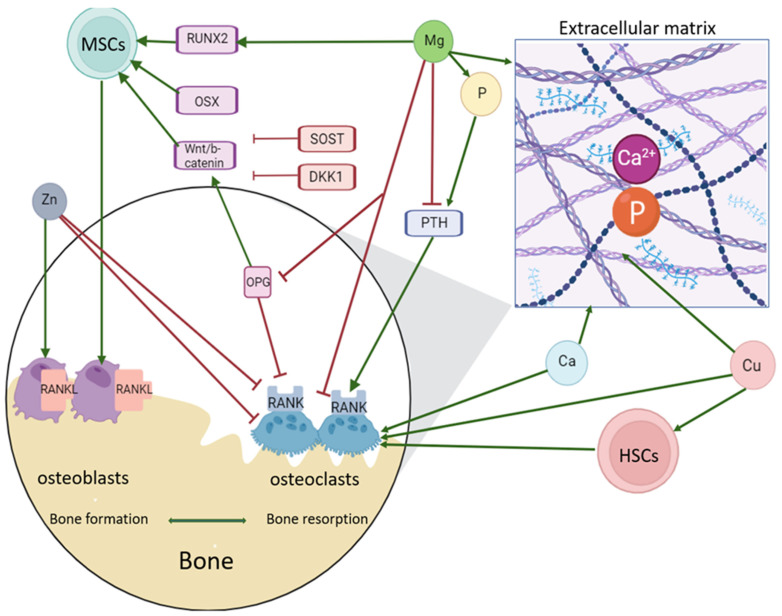
The influence of micronutrients on bone metabolism. OPG—osteoprotegerin, MSC—mesenchymal progenitor cell, RUNX2—Runt-related transcription factor 2, OSX—Osteoblast-specific Transcription Factor Osterix, WNT/b-catenin, SOST—Sclerostin gene, DKK1—Dickkopf WNT Signaling Pathway Inhibitor 1, RANKL—Tumor necrosis factor Superfamily Member 11 (Tumor necrosis factor ligand Superfamily Member 1), RANK, PTH—parathyroid hormone, HSC—hematopoietic progenitor cell, Mg—magnesium, Zn—zinc, Cu—copper, P—phosphorus, and Ca—calcium. Green lines—positive influence; red lines—negative influence.

**Table 1 diagnostics-15-00250-t001:** Characteristics of the studied groups.

	I Type OI	III Type OI	IV Type OI	V Type OI	Control
Total, N	29	8	5	3	45
Male, N (%)	14 (48.3)	3 (37.5)	2 (40.0)	1 (33.3)	17 (37.8)
Female, N (%)	15 (51.7)	5 (62.5)	3 (60.0)	2 (66.7)	28 (62.2)
Age, y.o.	32.0 [21.0; 38.0]	27.5 [23.5; 38.5]	29.5 [21.3; 37.0]	30.0 [28.0; 47.0]	27.0 [21.0; 35.0]
Height, sm	164.0 [151.0; 170.0] *p* = 0.027	108.5 [100.0; 127.5] *p* = 7.7 × 10^−6^	140.0 [139.0; 152.0] *p* = 6.9 × 10^−3^	136.0 [100.0; 140.0] *p* = 0.004	165.0 [162.0; 178.0]
Weight, kg	57.0 [51.0; 2.0]	36.5 [31.5; 40.5] *p* = 8.2 × 10^−5^	60.0 [53.0; 61.0]	43.0 [38.0; 60.0]	59.0 [53.0; 78.0]
BMI, kg/m^2^	24.0 [20.5; 26.8]	29.2 [26.7; 32.4] *p* = 5.4 × 10^−3^	24.0 [22.9; 30.6]	32.4 [21.9; 38] *p* = 0.036	21.2 [19.6; 24.3]
Fractures, N	11.0 [10.0; 20.0]	45.0 [30.0; 60.0]	20.0 [20.0; 40.0]	30.0 [15.0; 50.0]	0
Blue sclera, N (%)	27 (93.0)	8 (100.0)	4 (80.0)	2 (66.7)	0
HI, N (%)	11 (37.9)	1 (37.5)	2 (40.0)	0	0
JH, N (%)	21 (72.4) *p* = 5.6 × 10^−5^	2 (25.0)	2 (40.0)	1 (33.3)	10 (22.2)
DI, N (%)	17 (58.6) *p* = 7.9 × 10^−7^	8 (100.0) *p* = 4.2 × 10^−9^	4 (80.0) *p* = 2.6 × 10^−5^	3 (100.0) *p* = 1.9 × 10^−5^	2 (4.4)

Note: Data are presented as Me [Q1; Q3] and M ± sd. Me—median, Q1—first quartile, Q3—third quartile, M—mean, sd—standard deviation, N—total quantity, *p*—value in comparison with control group, HI—hearing impairment, JH—joint hypermobility, and DI—dentinogenesis imperfecta.

**Table 2 diagnostics-15-00250-t002:** Levels of micronutrients in serum in patients with osteogenesis imperfecta and the control group.

Micronutrients	Reference Values	OI, N = 45	Control, N = 45	*p*-Value
Calcium	2.02–2.55 mmol/L	2.45 [2.37; 2.55]	2.47 [2.4; 2.53]	0.488
Magnesium	0.66–1.07 mmol/L	0.86 ± 0.08	0.81 ± 0.06	5.4 × 10^−4^
Phosphorus	0.74–1.52 mmol/L	1.15 ± 0.24	1.31 ± 0.21	2.0 × 10^−3^
Zinc	10.7–19.5 µmol/L	13 [11.3; 14.5]	14 [12.2; 15.2]	0.045
Copper	11.0–24.4 µmol/L	19.9 [18.5; 21.6]	17.8 [16.8; 19.5]	6.9 × 10^−4^

Note: Data are presented as Me [Q1; Q3] and M ± sd. Me—median, Q1—first quartile, Q3—third quartile, M—mean, and sd—standard deviation.

**Table 3 diagnostics-15-00250-t003:** Comparative analysis of micronutrients of blood serum in patients with different types of osteogenesis imperfecta.

Micronutrients	I Type, N = 29	III Type, N = 8	IV Type, N = 5	V Type, N = 3	Control, N = 45
Calcium, mmol/L	2.48 [2.39; 2.56]	2.50 [2.28; 2.59]	2.40 [2.35; 2.40] *p* = 0.026	2.45 [2.36; 2.50]	2.47 [2.40; 2.53]
Magnesium, mmol/L	0.87 ± 0.086 *p* = 3.8 × 10^−4^	0.83 ± 0.06	0.85 ± 0.07	0.87 ± 0.09	0.81 ± 0.06
Phosphorus, mmol/L	1.16 ± 0.23 *p* = 0.01	1.18 ± 0.27	1.12 ± 0.26	0.98 ± 0.17 *p* = 0.014	1.31 ± 0.21
Zinc, µmol/L	13.10 [12.5; 14.6]	12.35 [11.0; 13.8] *p* = 0.048	11.4 [10.5; 13.2]	15.3 [12.0; 17.6]	14.0 [12.2; 15.2]
Copper, µmol/L	20.18 ± 3.38 *p* = 0.039	19.51 ± 1.91	21.80 ± 1.75 *p* = 0.026	18.80 ± 0.07	18.54 ± 3.09

Note: Data are presented as Me [Q1; Q3] and M ± sd. Me—median, Q1—first quartile, Q3—third quartile, M—mean, sd—standard deviation N—total quantity, and *p*—value in comparison with control group.

**Table 4 diagnostics-15-00250-t004:** Analysis of bone mineral density in the studied groups.

	I Type, N = 29	III Type, N = 8	IV Type, N = 5	V Type, N = 3	Control, N = 45
BMD (g/sm^2^)	1.03 [0.96; 1.11] *p* = 3 × 10^−5^	0.84 [0.75; 0.89] *p* = 1 × 10^−5^	0.97 [0.89; 1.11] *p* = 0.009	0.93 [0.79; 1.03] *p* = 0.006	1.182 [1.10; 1.22]
Z-score	−0.80 [−1.40; −0.10] *p* = 6 × 10^−9^	−2.15 [2.85; 0.35] *p* = 2 × 10^−4^	−0.60 [−1.50; 0.90]	−0.90 [−1.35; −0.75] *p* = 0.008	0.71 [0.08; 1.20]

Note: BMD—areal bone mineral density, N—total quantity, and *p*—value in comparison with control group.

## Data Availability

The data are contained within this article.

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
