# Peer review of "Complex Analysis of Micronutrient Levels and Bone Mineral Density in Patients with Different Types of Osteogenesis Imperfecta"

_diagnostics, 2025, doi:10.3390/diagnostics15030250_

Round 1
Reviewer 1 Report
Comments and Suggestions for Authors
Dear Author,
Thank you for the opportunity to review this article. It brings another light on OI regarding monitoring and treatment possibilities. However:
Line 21: "In the OI overall, magnesium and copper were elevated" could be revised to "In patients with OI, overall magnesium and copper levels were elevated". Several complicated expressions across the paper can be simplified for easiness of reading.
The introduction is thorough and well-grounded. However, it over-explains basic concepts of OI (e.g., prevalence) that could be condensed to make room for elaborating on micronutrient significance, as the literature contains several articles about this topic.
In the results, you should mention in each variable where the distribution was normal and allowed for T-tests and where it was not, implying the use of the Mann-Whitney test. Also, although the P-value is correctly expressed in tables, it is enough to write “p-value<0.05” for easier reading in the text paragraphs.
The Conclusions section is somewhat speculative. It would benefit from more direct linkage to the results. It must comprise 3-7 clear affirmations based on the results.
An interesting topic for further studies that may be mentioned in the Discussion is the relationship between micronutrients and the success rate of surgical treatment (I.E. Sofield osteotomies and IM rodding) because it represents an important part of OI's social and economic burden. Here is an article on this topic: 10.3390/children8111066.
Comments on the Quality of English LanguageA minor English revision is needed, regarding typographical errors I.E. “Lysine residuesin collagen” and word choice (“challenging to overstate” may be replaced with “Critical”).
Author Response
Dear reviewer! We are very grateful for the high assessment and review of the article, as well as for valuable comments and observations. We have tried to answer all the questions and make all the necessary corrections.
Specific comments include:
- Line 21: "In the OI overall, magnesium and copper were elevated" could be revised to "In patients with OI, overall magnesium and copper levels were elevated". Several complicated expressions across the paper can be simplified for easiness of reading.
Answer: Dear reviewer, we fully agree with the comment and edits have been made.
- Introduction:
- The introduction is thorough and well-grounded. However, it over-explains basic concepts of OI (e.g., prevalence) that could be condensed to make room for elaborating on micronutrient significance, as the literature contains several articles about this topic.
Answer: Dear reviewer, it was added to the Introduction:
«However, comprehensive understanding of the entire pathogenesis of OI is still lacking. Several studies have indicated that micronutrients are essential for the growth and development of the skeleton, as they interact with the micronutrients in bones and participate in normal bone metabolism [7]. It is known that certain micronutrients, such as phosphorus, calcium, copper, zinc, magnesium, vitamin D, and proteins, influence bone structure and metabolism, thereby preventing skeletal disorders [8–12]. However, there has been insufficient research examining these elements in patients with OI, predominantly focusing on pediatric cohorts. The precise roles of micronutrients in the pathogenesis of OI remain largely unknown, and understanding the relationship be-tween these elements and areal bone mineral density (BMD) may provide valuable in-sights for the diagnosis and intervention of OI. Moreover, the search for potentially new pathogenic targets for pharmacological therapy of OI represents a promising avenue for future research».
Results:
- In the results, you should mention in each variable where the distribution was normal and allowed for T-tests and where it was not, implying the use of the Mann-Whitney test. Also, although the P-value is correctly expressed in tables, it is enough to write “p-value<0.05” for easier reading in the text paragraphs.
Answer: Dear reviewer, it has been added to the "Materials and Methods" section
«Statistical analysis comparing two independent groups on parameters such as magnesium and phosphorus was conducted using Student's t-test. In the comparison of other parameters (age, height, weight, BMI, number of fractures, levels of calcium, zinc, and copper, BMD, z-score), the hypothesis of normality in distribution was rejected, and the Mann-Whitney U test was used».
Conclusions:
- The Conclusions section is somewhat speculative. It would benefit from more direct linkage to the results. It must comprise 3-7 clear affirmations based on the results.
Answer: Dear reviewer, it was added to the "conclusions" section:
« Despite serum concentrations of micronutrients remaining within reference values, differences were observed when comparing various types of OI and the control group. In Type I OI, higher concentrations of magnesium and copper were noted. Type V OI exhibited the lowest serum phosphorus levels. Type III OI was associated with lower zinc levels, and despite a higher number of fractures in this group, there was no significant difference compared to the control group in terms of calcium and phosphorus concentrations. Lower serum calcium levels were found exclusively in Type IV OI and did not correlate with DXA-derived BMD or the number of fractures in either the control or OI groups. The number of fractures demonstrated a moderate positive correlation with absolute BMD values, but not with the Z-score in the OI group.».
- Discussion:
- An interesting topic for further studies that may be mentioned in the Discussion is the relationship between micronutrients and the success rate of surgical treatment (I.E. Sofield osteotomies and IM rodding) because it represents an important part of OI's social and economic burden. Here is an article on this topic: 10.3390/children8111066.
Answer: Dear reviewer, was added to the «Discussion»;
«Currently, bisphosphonate therapy and surgery utilizing telescopic rods are considered the gold standard for the treatment and correction of bone deformities in OI [41]. The use of micronutrients has been proposed as a therapeutic alternative to enhance bone healing following the implantation of metal constructs. Researchers from Brazil investigated the impact of adding micronutrients (calcium, magnesium, zinc) on the restoration of bone tissue around implants in mice. No statistically significant differences were observed between the groups receiving placebo and those receiving micronutrients supplementation based on computed tomography results and the measurement of the torque required to disrupt the bone-implant interface [42]. The effects of other micronutrients on the recovery of bone tissue following surgical intervention warrant further investigation.»
Comments on the Quality of English Language:
- A minor English revision is needed, regarding typographical errors I.E. “Lysine residuesin collagen” and word choice (“challenging to overstate” may be replaced with “Critical”).
Answer: Dear reviewer, we fully agree with the comment and have made the necessary changes.

Reviewer 2 Report
Comments and Suggestions for Authors
Dear Authors, Thank you for the opportunity to review your research paper on a complex analysis of micronutrient levels and bone mineral density in patients with different types of Osteogenesis Imperfecta. Osteogenesis imperfecta is a rare but serious disease that greatly reduces the quality of life of patients. However, the pathogenesis of this lesion is still unknown. This study focuses on the analysis of micronutrient levels, which has contributed significantly to the elucidation of this unknown lesion. This work is also a preliminary study that fills an evidence gap. However, the following improvements would further increase the robustness of the paper.
Introduction
P2L63
The background of Osteogenesis Imperfecta is well described in this paper. However, the rationale for the focus on micronutrients is weak. In addition, the authors' hypothesis, which should be clarified, should be stated before the purpose of the study.
Material and Methods
P2L66
This study design is a case-control study, not a cross-sectional study. Case-control studies are appropriate for the analysis of rare diseases such as Osteogenesis Imperfecta, which is why this study has a properly prepared control group.
Inclusion and exclusion criteria are not sufficiently explained. Authors should explain each criterion in detail with a subtitle.
P2L74
Was it one doctor who diagnosed Osteogenesis Imperfecta? Or several doctors?
P2L79
A new subtitle for statistical analysis should be created.
P2L82
Authors should include Excel and GraphPad versions, product information, company, city, and country.
Results
P3L107
This is the first mention of a control group. The selection criteria for the control group should be addressed in the Materials and Methods section, not in this section.
Figure 1
Can we interpret this figure to mean that OI type 1 and healthy individuals have approximate BMD medians and are difficult to diagnose clinically? If so, it would be better to highlight it in a figure legend or discussion section.
Discussion
・There is a detailed description of the micronutrients, which the reader will understand well. However, the present results indicate that all were within the normal range, although there were some differences by species. More explanation and discussion of this phenomenon should be added.
・The authors need to address the limitations of this study.
Author Response
Dear reviewer, thank you for your high appreciation of the article and comprehensive analysis of the article. Your valuable comments will help us improve the quality of the article. The team of authors will try to respond to all your comments.
Introduction
P2L63
The background of Osteogenesis Imperfecta is well described in this paper. However, the rationale for the focus on micronutrients is weak. In addition, the authors' hypothesis, which should be clarified, should be stated before the purpose of the study.
Answer: Dear reviewer, it was added to the Introduction
“ However, comprehensive understanding of the entire pathogenesis of OI is still lack-ing. Several studies have indicated that micronutrients are essential for the growth and development of the skeleton, as they interact with the micronutrients in bones and participate in normal bone metabolism [7]. It is known that certain micronutrients, such as phosphorus, calcium, copper, zinc, magnesium, vitamin D, and proteins, influence bone structure and metabolism, thereby preventing skeletal disorders [8–12]. However, there has been insufficient research examining these elements in patients with OI, predominantly focusing on pediatric cohorts. The precise roles of micronutrients in the pathogenesis of OI remain largely unknown, and understanding the relationship be-tween these elements and areal bone mineral density (BMD) may provide valuable in-sights for the diagnosis and intervention of OI. Moreover, the search for potentially new pathogenic targets for pharmacological therapy of OI represents a promising avenue for future research”.
Мaterial and Methods
P2L66
This study design is a case-control study, not a cross-sectional study. Case-control studies are appropriate for the analysis of rare diseases such as Osteogenesis Imperfecta, which is why this study has a properly prepared control group.
Answer: Dear reviewer, the authors agree with your comment. Corrected to "case-control".
Inclusion and exclusion criteria are not sufficiently explained. Authors should explain each criterion in detail with a subtitle.
Answer: Dear Reviewer, the authors fully agree with the remark. A section titled "Control Group" has been added to the Materials and Methods section.
"The inclusion criteria were: young age from 18 to 35 years.
Exclusion criteria included: presence of hereditary forms of connective tissue disorders (Marfan syndrome, Ehlers-Danlos syndrome, and osteogenesis imperfecta), metabolic bone diseases, autoimmune diseases, history of trauma, chronic kidney disease stage 3b or higher, presence of endocrine diseases, conditions associated with micronutrient metabolism, pregnancy and lactation, taking medications that affect bone mineral density, and refusal to participate."
P2L74
Was it one doctor who diagnosed Osteogenesis Imperfecta? Or several doctors?
Answer: Dear Reviewer, a sentence has been added to the "Materials and Methods" section: " The diagnosis of OI was established based on clinical presentation and molecular genetic diagnostics at the connective tissue research center of the BSMU clinic by multidisciplinary team that consisted of a rheumatologist, a general practitioner, and a medical geneticist."
P2L79
A new subtitle for statistical analysis should be created.
Answer: Dear reviewer, the subtitle " Statistical data processing " has been added
P2L82
Authors should include Excel and GraphPad versions, product information, company, city, and country.
Answer: Added «GraphPad Prism 10.4.0.(GraphPad Software, San Diego, California, USA), Excel 2021, Version 2109 (Microsoft Office 2021, Microsoft Corporation, USA)».
Results
P3L107
This is the first mention of a control group. The selection criteria for the control group should be addressed in the Materials and Methods section, not in this section.
Answer: Dear Reviewer, the fragment "The control group included 17 (37.8%) men and 28 (62.2%) women, aged 20 to 39 years (median age of 27.0 years), comparable in gender composition and age to the OI patient group" has been relocated to the subsection "Control Group" in the Materials and Methods.
Figure 1
Can we interpret this figure to mean that OI type 1 and healthy individuals have approximate BMD medians and are difficult to diagnose clinically? If so, it would be better to highlight it in a figure legend or discussion section.
Answer: Dear Reviewer, diagnosis of OI usually establishes by main clinical manifestations like recurrent fractures. Reduced bone mineral density is not a criterion for the diagnosis of osteogenesis imperfecta, since BMD evaluates bone mineralization, not architectonics. But for better understanding the fragment was added “obtained value of BMD for Type I OI is close to those of the control when compared to other types of OI, yet they still differ significantly, with a high level of significance (1.182 g/sm² and 1.03 g/sm², p<0.05)”
Discussion
・There is a detailed description of the micronutrients, which the reader will understand well. However, the present results indicate that all were within the normal range, although there were some differences by species. More explanation and discussion of this phenomenon should be added.
Answer: Dear Reviewer, the following has been added to the "Discussion" section:
"The level of zinc in the blood was positively associated with lumbar BMD (total T-score) among postmenopausal women in Turkey. Suzuki T. et al. (2015) suggested that zinc deficiency may indirectly influence BMD by reducing intestinal calcium absorption and enhancing the effects of bone resorption by parathyroid hormone [23] due to elevated levels of parathyroid hormone [24]. It is worth noting that the use of these micronutrient reference values may not be entirely appropriate in the case of rare diseases such as OI, and therefore, the development of specific micronutrient standards for patients with this disease may be necessary".
・The authors need to address the limitations of this study.
Answer: Dear Reviewer, information regarding the limitations has been added to the "Discussion" section:
"Our study has several limitations. Only 45 patients from the primary group were included in the study of the rare condition, necessitating an expansion of the sample size. Additionally, our research does not account for comorbidities in patients with the condition. Furthermore, our study is of the case-control type, which does not allow for a comprehensive assessment of the causal relationships between the components of the pathogenesis and the levels of micronutrients. It is also important to note that only serum levels of micronutrients were assessed; to obtain a complete picture, it is necessary to evaluate the concentrations of these micronutrients directly in bone tissue."

Round 2
Reviewer 1 Report
Comments and Suggestions for Authors
The authors have answered my requests to improve the paper.
Reviewer 2 Report
Comments and Suggestions for Authors
All modifications confirmed.